# Physical Activity Counseling among Adults in Primary Health Care Centers in Brazil

**DOI:** 10.3390/ijerph18105079

**Published:** 2021-05-11

**Authors:** Letícia Pechnicki dos Santos, Alice Tatiane da Silva, Cassiano Ricardo Rech, Rogério César Fermino

**Affiliations:** 1Research Group in Environment, Physical Activity and Health, Federal University of Technology—Parana, Curitiba 81310-900, Brazil; letynicki@hotmail.com (L.P.S.); silva.alice@outlook.com (A.T.S.); 2Postgraduate Program in Physical Education, Federal University of Parana, Curitiba 81531-980, Brazil; 3Physical Education Department, Federal University of Santa Catarina, Florianopolis 88040-900, Brazil; cassiano.rech@ufsc.br

**Keywords:** directive counseling, walking, motor activity, exercise, primary care, health services accessibility, public health, epidemiologic studies

## Abstract

Physical activity (PA) counseling by health professionals has promising results in behavior change. However, few studies have evaluated its prevalence in Primary Health Care in Latin American countries. This study aimed to describe the prevalence and analyze the associated factors of PA counseling in adults in Primary Health Care in Brazil. This is a cross-sectional study with a representative sample of 779 adults (70% women). Counseling was identified among those who reported having received PA counseling during a health professional consultation in the last 12 months. Sociodemographic factors, health conditions, and leisure-time PA were analyzed with Poisson regression. The prevalence of counseling was 43% (95% Confidence Interval [CI]: 39.5–46.4%), higher in people aged ≥40 years (Prevalence Ratio [PR]: 1.44; 95% CI: 1.19–1.75], who are married (PR: 1.27; 95% CI: 1.07–1.59), obese (PR: 1.53; 95% CI: 1.23–1.90), take prescription medication (PR: 1.83; 95% CI: 1.47–2.27), and walk for leisure (PR: 1.28; 95% CI: 1.06–1.54). People with more education were less likely to receive PA counseling (PR: 0.82; 95% CI: 0.68–0.99). In conclusion, 4 out of 10 users reported receiving PA counseling and this was associated with sociodemographic factors, health conditions, and walking for leisure. These results can guide PA promotion in Primary Health Care.

## 1. Introduction

In most countries, the current health scenario is a consequence of an increase in chronic diseases due to the adoption of unhealthy lifestyles, emphasizing physical inactivity [1]. Despite the benefits of regular physical activity (PA), about 28% of the world adult population does not follow the World Health Organization (WHO) guidelines [2,3]. Physical inactivity prevalence is high among Latin American countries (39%), especially in Brazil (47%) [2]. Besides being a risk factor for developing chronic diseases, physical inactivity negatively affects mental and physical health and quality of life [1,3].

Thus, integrated programs of PA promotion at the population level have been implemented in many countries globally, which presuppose the interrelationship between different community sectors, such as school, active commuting, built environment, and health professionals [4,5,6]. In this context, the Brazilian Primary Care Policy was approved in 2011 [7] and aims to promote health through actions described as Basic Health Units (BHUs—Unidades Básicas de Saúde, UBS in Portuguese), as these are the gateway for the population to the Unified Health System (Sistema Único de Saúde, SUS in Portuguese) [7,8,9]. It becomes relevant to incorporate these health counseling actions, raising awareness about the adoption of healthy behaviors [5,10,11,12]. Conceptually, counseling is an educational practice by health professionals to empower individuals to become active as part of their overall health maintenance [10,13,14,15,16,17]. This practice respects people’s autonomy and values their potential, thus promoting behavioral changes and improving their quality of life [10,11,12].

In this context, all health professionals should counsel or guide their patients about PA [10,11], and consultations represent an essential opportunity for this educational practice, due to the high confidence in these professionals, especially in physicians [18]. In Brazil, 76% of adults (≈160 million) consulted with a physician in the past 12 months [8]. Moreover, 70% of the population regularly uses public health services, with BHUs being the most reported (47%). Brazil has 42,488 BHUs, and the purpose of these health centers is to deal with up to 80% of the population’s health problems without hospitals. Women, middle-aged adults and the elderly, people of low-economic status, people of color, and married are the most frequent users of BHUs [8].

PA counseling by health professionals has had promising results in behavior change and is recommended as part of integrated community interventions [5,11,12,16,17]. The WHO, the American College of Sports Medicine, and the International Society for Physical Activity and Health have recommended that health professionals promote PA through counseling [3,4,19,20]. Besides being low-cost, easy to apply, and easily used by health professionals [13,14,15], evidence shows a positive effect of counseling on PA levels [11,12,16,17].

Patient PA counseling is underutilized [20], and few studies have evaluated this action in Primary Health Care in Latin American countries [21]. The prevalence is low (37%) and even lower in high-income countries (34%) when compared to others (43%) [21]. In Brazil, the research has investigated the PA counseling received by adults [22,23,24,25,26,27,28,29] and the elderly [22,23,25,26,27,28,29,30]. While some studies explored a representative population-based sample [23,24,30] and residents of the coverage area of BHUs [22], others were explicitly conducted in BHUs [22,25,26,27,28,29]. In these studies, the prevalence of PA counseling received by BHUs users was 42%, and many sociodemographic factors (sex, age, marital status, skin color, education, economic level) and health conditions (overweight, hypertension, diabetes, dyslipidemia) were analyzed and show inconsistency and contradictory results. For example, while some studies show an absence of association between sex and counseling [25,26,28], others found a positive association with the female sex [22,26], while others found an inverse association [27]. Only one study in BHUs analyzed the association between counseling and leisure-time PA (LTPA) [25]. Nevertheless, some studies show high-risk bias [21] (e.g., PA counseling evaluated in combination with counseling for healthy eating habits [27], sample not representative, participants intentionally selected from few BHUs [27,28,29]), which may not represent the populational sociodemographic and the contextual reality of specific places and cities.

It is essential to expand this knowledge, based on robust studies with a representative sample from BHUs in order to identify the individuals who are more exposed to PA counseling. This approach and its results are essential so that health professionals can implement and direct specific actions to promote PA towards disadvantaged population groups and those facing barriers to starting PA, whether due to psychosocial issues or lack of information, based on each city’s context or region. Thus, this study aimed to describe the prevalence and to analyze the associated factors of PA counseling in adults in Primary Health Care centers in Brazil.

## 2. Materials and Methods

### 2.1. Study Characteristics and Ethical Aspects

This study was reported according to the Strengthening the Reporting of Observational Studies in Epidemiology (STROBE) guidelines. In 2019, a quantitative, observational, and cross-sectional study was conducted in a representative sample of adults in the BHUs of an urban area of São José dos Pinhais (Parana state, Southern Brazil). The general project aimed to evaluate PA or sedentary behavior counseling, LTPA, and some of their determinants [31].

BHUs are public clinics strategically distributed within a city, with free access for Primary Health Care provided by physicians, nurses, pharmacists, physiotherapists, nutritionists, psychologists, and community health agents [9]. In some Brazilian cities, some BHUs also include physical education professionals [32,33].

São José dos Pinhais is a developed city in the metropolitan area of Curitiba (state capital), a distance of 18 km from downtown to downtown. It has an area of 946 km^2^ (21% urban), there is an estimated population of 329,058 inhabitants (medium size), a population density of 280 inhabitants km^2^, and a high Human Development Index (0.758). Some indicators among São José dos Pinhais, Curitiba, and Brazil are similar (Table 1) [34].

The city has 413 health establishments, among which are 27 BHUs (56% in the urban area). Only BHUs in the urban area (*n* = 15) were intentionally selected, as these are accessible to 90% of the city’s population. The study was approved by the National Commission for Ethics in Research (CONEP) of the National Health Council, with the Certificate of Presentation of Ethical Appreciation (CAAE) under protocol number 95985118.0.0000.0020, and by the Research Ethics Committee of the Pontifical Catholic University of Parana under the number 2.882.260. Participants were consulted, informed about voluntariness, and agreed to participate in the research by signing an informed consent form, according to the recommendations of CONEP.

### 2.2. Sample Size, Number of Participants, and Sample Power

The number of participants was estimated based on the average number of visits made in each BHU between January and February 2019 (*N* = 34,275). For sample representation, we considered the prevalence of PA counseling received in BHU (30%—based on a literature review available during the project planning and pilot study) and a 95% confidence level. A sampling error of four percentage points and a design effect of 1.5 were also regarded [35]. As a result, the minimum number of participants was estimated at 745 people. However, with a 10% increase for losses and refusals, we estimated a minimum of 820 adults. Nevertheless, it was decided to approach a surplus of 100 people (*n* = 920) to reduce the estimation errors for multivariate analyses in future studies. The sample size was proportionally calculated by the number of visits to each BHU and varied between 31 to 92 users.

A total of 779 participants were interviewed (544 women and 235 men). A posteriori statistical analyses showed that, based on the predictor variables, our sample had an average power of 90% (β: 20%) and a confidence level of 95% (α: 5%) to detect prevalence ratio (PR), identified as significant in our study. This analysis was conducted using software G*Power (University of Duesseldorf, Duesseldorf, Germany).

### 2.3. Selection of Participants

The participants were systematically selected based on their position in the waiting room at the BHU, counting between the first and fifth, from left to right of the entrance door [25]. The third user was selected, approached, and invited to participate in the study. In case of refusal, if the participant did not meet the inclusion criteria, the first person on the left was then selected. However, in some BHUs, this procedure was unfeasible due to the small number of people present and differences in the waiting rooms’ layout. In these cases, the first users to the left were approached.

Only adults (≥18 years) were eligible and invited. Among them, we excluded people who lived outside the urban area, and were using the BHU for the first time, had some physical limitation for PA practice (e.g., wheelchair and crutch users), or had some cognitive or phonation limitation that prevented comprehension of the questionnaire (e.g., hearing impairment, mental disorders) (*n* = 9).

### 2.4. Data Collection

Ten trained interviewers conducted face-to-face interviews before or after consultation by health professionals in an individual, reserved room to ensure no external influence on the responses. The average time of application of the questionnaire was 18 min (±5 min, 9–55 min). The data was collected between April and October 2019.

### 2.5. Variables

#### 2.5.1. Outcome Variable: Physical Activity Counseling

PA counseling was evaluated based on dichotomous response to the question: “During the last year (12 months), any time you were at the BHU, did you receive PA counseling during the consultation by a health professional (advice, tips, or guidance on PA to change/improve your health)?” (no, yes). This measure has been used in similar studies and adapted to the local context [21,22,23,24,25,26,27,28,29,30].

#### 2.5.2. Predictor Variables

Based on the literature review, predictive variables included sociodemographic factors (sex, age group, marital status, skin color, education, and economic level), health conditions (Body Mass Index—BMI, smoking, chronic disease diagnosis, medications consumption), and LTPA [16,21,22,23,24,25,26,27,28,29,30,36]. These variables are described below.

##### Sociodemographic Characteristics

The sex was observed (male, female), and the participants grouped into three age groups (young adults: 18–39 years, middle-aged adults: 40–59 years, and elderly: ≥60 years old). The marital status was evaluated into three categories (single, married or stable relationship, divorced or widowed) and grouped into two (single, divorced, widowed or married, stable relationship). Skin color was self-reported into five categories (white, black, yellow, brown, and indigenous) and categorized as white and non-white (other categories). Education was evaluated into five levels (illiterate or incomplete elementary I, complete elementary I or incomplete elementary II, complete elementary II or incomplete high school, complete high school or incomplete university and complete university), and then regrouped into three categories: elementary school or less, junior high school, high school or more. All variables were measured according to the method proposed by the Brazilian Institute of Geography and Statistics and adopted by the Brazilian Health Surveillance System [37].

Socioeconomic status (SES) was evaluated by a standardized questionnaire [38] which considers ownership of domestic appliances, education level of the family head, having a housekeeper, and public services (water and paved street), and participants were classified into seven levels (A1–higher, A2, B1, B2, C, D, E). However, these categories were grouped into low economic level (SES C + D + E) and high (SES A + B) for analysis purposes.

##### Health Conditions

The BMI was calculated using self-reported weight and height data and classified into three categories (low or normal weight: ≤24.9 kg/m^2^, overweight: 25–29.9 kg/m^2^, obese: ≥30.0 kg/m^2^) [39]. Smoking habit was evaluated by a dichotomous response (yes, no), while chronic diseases were identified by a self-reported medical diagnosis of hypertension, diabetes, dyslipidemia, or coronary artery disease. In this regard, interviewees were classified into three groups due to the number of chronic diseases (0, 1, ≥2). Finally, respondents reported the continuous use of medication for chronic diseases. In this sense, they were classified into three categories according to the number of medications (0, 1–3, ≥4). These measures are valid and based on the Brazilian Health Surveillance System [37].

##### Leisure-Time Physical Activity (LTPA)

The PA in a usual week was measured with the leisure module of the long version of the International Physical Activity Questionnaire (IPAQ) [40,41,42]. Participants self-reported the weekly frequency and mean daily volume of walking, moderate, and vigorous PA. The score in each activity/intensity was obtained in minutes per week (min/week) by multiplying the weekly frequency by mean daily volume. Total LTPA was obtained summing min/week of walking + min/week of moderate PA + (min/week of vigorous PA × 2) [41]. Thereby, walking and total LTPA were classified according to WHO guidelines (≥150 min/week) [3].

### 2.6. Data Quality Control

Quality control of data was assured through six steps. First, all interviewers (members of the research group and undergraduates and students from the Postgraduate Program in Physical Education) received a 20 h theoretical and practical training on technical procedures for interviewing (approaching participants, recording losses and refusals, application of questionnaires, and coding forms) based on an instruction manual prepared by the core project team. The interviewers strictly followed all these procedures; they were blinded to the investigation’s objectives and hypotheses and supervised by a field coordinator. Second, a pilot study was conducted on a random sample of 81 participants from three BHUs to test the data collection procedures and the comprehension of the questions translated from other studies and adapted to the local context. Third, all pilot study participants were re-interviewed at an interval of 7 to 10 days to analyze the temporal stability of the main variables of the study. Temporal stability (reproducibility) of PA counseling was analyzed with percentage agreement and Cohen’s kappa test, which showed 88% agreement and good Kappa value (0.77; *p* < 0.001). Fourth, data entry was conducted by the field coordinator using the EpiData software (EpiData Association, Odense, Denmark). Fifth, data was cleaned using exploratory analysis, with SPSS software (v. 26.0, IBM SPSS Statistics, Armonk, NY, USA), to identify possible typing errors in data entry for each variable, to detect outliers, and to verify all variables’ distributions. Finally, each variable outlier was personally checked in the questionnaire and manually corrected in the database.

### 2.7. Statistical Analysis

Descriptive analyses were performed using frequency and percentage distributions. The prevalence of PA counseling was described among predictor variables and their associations analyzed by Poisson regression (Prevalence Ratio [PR] and 95% of confidence interval [95% CI]). Initially, variance inflation factor (VIF) tests were performed and rejected the hypothesis of multicollinearity (1/VIF ≥ 0.71). After bivariate analysis, all variables at the same or higher level, which presented a *p*-value < 0.20, were selected for adjustment by multivariate model. Thus, the final model was conducted following a multiple model analysis, which was elaborated from a hierarchical structure with the following levels and variables: level 1: sociodemographic characteristics; level 2: health conditions; and level 3: LTPA [43]. All analyses were performed using the STATA software (v. 16, StataCorp, College Station, TX, USA) and at a 5% significance level. The correction for design effect was performed using the command *svy* to account the estimates of outcome variability.

## 3. Results

We approached 935 users, the refusal rate was 14% (*n* = 134), and losses 2% (*n* = 22), which resulted in 779 participants interviewed. Most participants were women (69.8%), aged between 18–39 years (45.2%), married (64.0%), identified as having white skin color (73.0%), with education level of high school or higher (50.4%), and low economic level (71.2%) (Table 2).

Regarding health conditions, 67.3% of participants were overweight or obese (BMI ≥ 25 kg/m^2^), 14.9% smokers, 35.9% hypertensive, 15.7% diabetics, 15.9% dyslipidemic, and 6.5% had received a medical diagnosis of coronary artery disease. Just over half of the participants reported zero chronic diseases (54.8%) and consumed one or more prescribed medications (51.3%) (Table 2). Walking and total LTPA ≥150 min/week were reported by 13.4% and 24.8% of participants, respectively. The prevalence of PA counseling was 43.0% (95% CI: 39.5–46.4%) (Table 2).

In the bivariate analysis (Table 3), the sociodemographic characteristics age group (40–59 years, PR: 1.53; 95% CI: 1.26–1.84|≥60 years, PR: 1.60; 95% CI: 1.29–1.99), marital status (married, PR: 1.29; 95% CI: 1.08–1.55), and education (junior high school, PR: 0.73; 95% CI: 0.57–0.93|complete high school or more, PR: 0.72; 95% CI: 0.60–0.85) were associated with PA counseling. Regarding health conditions, except for smoking (PR: 0.78; 95% CI: 0.59–1.01), all the others showed a positive association with counseling (*p* < 0.05). Walking (PR: 1.45; 95% CI: 1.20–1.73) and total LTPA (PR: 1.22; 95% CI: 1.02–1.44) showed a positive association with a higher prevalence of PA counseling.

After adjusting for possible confounding parameters (Table 3), the highest prevalence of counseling was observed among those aged 40–59 years (PR: 1.44; 95% CI: 1.19–1.75) and ≥60 years (PR: 1.44; 95% CI: 1.14–1.83) and for those married (PR: 1.27; 95% CI: 1.07–1.59). However, the education level complete high school or more showed an inverse association with the outcome (PR: 0.82; 95% CI: 0.68–0.99). Regarding health conditions, people with BMI ≥30.0 kg/m^2^ (PR: 1.53; 95% CI: 1.23–1.90) and those who take continuous use medication (1–3 medications, PR: 1.83; 95% CI: 1.47–2.27|≥4 medications, PR: 1.66; 95% CI: 1.21–2.28) were more likely to receive PA counseling. Finally, leisure walking was positively associated with PA counseling received by health professionals (PR: 1.28; 95% CI: 1.06–1.54).

## 4. Discussion

Our study aimed to explore the possible factors associated with PA counseling received by adults in BHUs of Primary Health Care in a medium-sized city in southern Brazil. With low risk of bias, the method used allowed us to represent the adult users of BHUs in an urban area, besides exploring the main individual predictors of counseling, analyzing the association’s strength and direction. Also, the variables were measured by standardized procedures and instruments, and these are our study’s strengths. Counseling was higher among middle-aged adults and the elderly, those with less education, married, obese, those who consume medications for chronic diseases, and those who walk in leisure-time. Studies have shown a positive effect of counseling received and PA levels [12,16,17]. Thus, our findings can support specific actions to improve Primary Health Care counseling in Brazil, specifically in BHUs [44,45]. However, PA counseling could be directed towards disadvantaged and less physically active population groups, such as women, the elderly, married, those with less education, those at a low economic level, obese, and users with chronic disease.

Our results showed that the participants had worse health conditions when compared to adults in the capital Curitiba, a city next to São José dos Pinhais (e.g., overweight: 54%, smoking: 11%, hypertension: 21%, diabetes: 7%) [37]. In part, this profile can be explained by characteristics inherent to this population subgroup and by the purpose of seeking medical consultation in BHUs due to current health conditions [8].

About 4 out of 10 users reported receiving PA counseling (43%). A systematic review synthesized Primary Health Care results in several countries and showed an average prevalence of 37% (±14%) [21]. However, in Brazil, population-based studies show average general counseling of 48% (±16%) [23,24,30]. In studies conducted exclusively with BHU users, the prevalence is 42% (±11%) [22,25,26,27,28,29]. The low prevalence found in BHUs can be explained by possible forgetfulness among respondents regarding the received counseling, the structural characteristics of Primary Health Care in the studies, perception of barriers between users of counseling services and health professionals, and lack of protocols for conducting counseling, which could result in different perceptions by users about what was advised [21]. For example, users may not understand health professionals’ guidance in BHUs as an orientation, but merely as an informal recommendation without a technical basis [46].

In our study, middle-aged and elderly adults received more counseling compared to young adults. Many studies have explored these variables, showing consistent results of higher prevalence of PA counseling according to age [22,23,24,25,27,46,47,48]. A higher occurrence of chronic diseases and low PA level can partially explain the greater amount of counseling with advancing age, which could cause health professionals to advise these users to be physically active [24,26,46]. These results are positive, adequate, and expected, since physical inactivity is positively associated with age [2]. However, two studies showed an inverse association between age and counseling [26,49]. The authors justify the possible inconsistency of the association as due to the possible lower confidence that health professionals have in encouraging changed behavior in older people, related PA [26]. In this context, the difference in results between studies suggests that more research is needed to expand knowledge about the relationship between counseling and age [26].

The married status was positively associated with receiving counseling. A few studies have explored this variable and found contradictory associations [22,24,25]. Our results are similar to those found in two studies [22,24], but the authors do not explain the possible causal relationship among these variables. In only one study, the authors justified this association because married individuals are generally older, have chronic diseases, and attend more medical consultations, which might have increased the likelihood of receiving counseling compared to single people [24]. However, regardless of the possible “causal relationship”, since Brazilian married individuals are less physically active in leisure-time [50], the greater amount of counseling among this population subgroup is important and could stimulate PA behavior change.

Regarding education, people who completed high school or more were less likely to receive counseling. These results are not well understood in the literature, as studies have shown that higher education levels were positively associated with counseling [26,36,47]. According to the authors, people with higher education could be more likely to follow counseling properly [47], while individuals with less education would have more difficulty understanding and putting advice into practice [26]. Moreover, some studies have shown an inverse association between counseling and education. Schooling is an essential sociodemographic factor in reducing health care disparities related to PA counseling in Primary Health Care among BHUs users.

The consumption of a high number of medications showed a positive association with counseling. This result is similar to other studies found in the literature [22]. In some of these, the authors report that people who consume more medication often have more health problems, needing more counseling to start or increase PA [22,51]. The authors have suggested that patients consuming lots of medication need more attention and health care and, as a result, receive more counseling from health professionals [51]. We also found that obesity was positively associated with receiving counseling. This is consistent and similar to other studies [24,46,48]. This association can, in part, be explained by the fact that obesity is one of the main risk factors for chronic diseases and because obese people have a negative perception of health, causing health professionals at BHUs to counsel more often [52]. Such a link should alert health professionals to advise people whose needs are not easily visible, whereas this practice is essential for everyone [46].

The positive association between counseling and walking can be explained by the recurrent counseling offered in Primary Health Care [53], especially in Brazil [23,33,54]. Health professionals usually advise their patients to walk for leisure [23,51,53]. Some authors have suggested that leisure is PA domain that can be substantially altered by counseling interventions when compared, for example, to the occupational domain [44]. These results can also be partially explained by the lack of knowledge of health professionals in BHUs on recommendations in other PA domains [13]. This is likely due to professionals’ low confidence in advising other PA types for their patients (exercises for muscle strengthening, running, cycling). This characteristic highlights the importance of having a physical education professional in Primary Health Care units as part of the health team [14,32,33]. Thereby, people whom other health professionals had instructed to start or increase PA could seek a specialist during their consultations at BHUs, thus receiving detailed information on other possible activities to be performed (e.g., domains and types of PA, frequency, intensity, daily or weekly volume, progression) [20]. Alternatively, this could reduce time spent in sedentary behavior at home, work, and daily commuting [24,32]. Analyzing the association between counseling and PA is relevant since many people Brazilian populations do not have enough financial resources to cover tuition for personalized guidance services and private establishments, such as clubs and gyms, for PA [25,55]. Thus, adequate physician guidance could, for example, encourage BHU users to practice PA in public open spaces in the neighborhood (e.g., walking or bike paths, fitness zones in urban parks) [4,5].

Contrary to what was expected, some predictors (e.g., sex, economic level, chronic diseases) had no association with counseling received. However, other studies have pointed to a positive association between some of these factors and counseling [24,49]. Such results can be positively interpreted because, although individuals with disease need differentiated care, users receive counseling equally regardless of diseases [51]. According to Primary Health Care principles in Brazil, one should not seek only curative actions, but also preventive health promotion actions with universality, integrality, equity, etc. [7,9].

Our results must be interpreted considering some limitations. The sample was restricted to adult BHU users from an urban area of a medium-sized city in southern Brazil. This city has a broad rural area (79% of the territory), comprising 12 BHUs. Thus, our findings cannot be extrapolated to other contexts (e.g., rural areas, larger cities, other states or regions of Brazil, and other countries). There were no physical education professionals in the health teams or PA programs of the selected BHUs. These may, in part, have altered the prevalence of counseling received, as well as associations between counseling and PA [33,54]. Finally, the cross-sectional design limits establishment of causality among the predictors and analyzed outcome.

## 5. Conclusions

The prevalence of PA counseling was 43%, and the populational groups more exposed to counseling were middle-aged and elderly adults, married, those with less education, obese, those who consume medications and walk in leisure-time.

Primary Health Care actions in Brazilian BHUs should be expanded, with health professionals’ training towards promoting PA, emphasizing epidemiology, public health, and PA in addition to developing counseling methods in routine service, with valid and useful protocols to evaluate actions’ effectiveness [19]. Health professionals should also know about public open spaces and structures for PA near the BHUs to recommend patients practice PA in these places [4,5]. Future studies should evaluate the effectiveness of different counseling actions regarding PA levels for different population groups due to differences in lifestyle between urban and rural areas, large and small cities, different regions in Brazil, and other countries. It is just as essential to develop PA counseling routines and protocols that different health professionals can use.

## Figures and Tables

**Table 1 ijerph-18-05079-t001:** Socioeconomic, health-related, and demographic indicators of São José dos Pinhais, Curitiba, and Brazil.

Variables	São José dos Pinhais	Curitiba	Brazil
Estimated population in 2020	329,058	1,948,626	211,755,692
Women	51%	52%	51%
Child mortality	10.6%	11.9%	16.7%
Illiteracy (≥15 years)	3.4%	2.1%	9.6%
Schooling (6–14 years)	97.4%	97.6%	99.3%
Human Development Index	0.758	0.823	0.727
Per capita income (BRL)	846.9	1581.0	767.0
Gini index per capita	0.4599	0.5652	0.6086

**Table 2 ijerph-18-05079-t002:** Descriptive characteristics of adults in Primary Health Care units. São José dos Pinhais, Parana, Southern Brazil, 2019 (*n* = 779).

Variable	Category	*n*	%
*Sociodemographic characteristics*			
Sex	Female	544	69.8
	Male	235	30.2
Age group (years)	18–39	346	45.2
	40–59	283	36.9
	≥60	137	17.9
Marital status	Single	280	36.0
	Married	497	64.0
Skin color	White	566	73.0
	Non-white	209	27.0
Education level	Elementary education or less	247	31.7
	Junior high school	139	17.8
	High school or more	393	50.4
Economic level	Low	555	71.2
	High	224	28.8
*Health conditions*			
Body mass index (kg/m^2^)	<24.9	248	32.2
	25–29.9	291	37.4
	≥30.0	230	29.9
Smoking	No	660	85.1
	Yes	116	14.9
Hypertension	No	499	64.1
	Yes	280	35.9
Diabetes	No	657	84.3
	Yes	122	15.7
Dyslipidemia	No	655	84.1
	Yes	124	15.9
Coronary artery disease	No	728	93.5
	Yes	51	6.5
Number of chronic diseases	0	427	54.8
	1	204	26.2
	≥2	148	19.0
Number of prescribed medications	0	387	49.7
	1–3	277	35.6
	≥4	115	14.8
*Leisure-time physical activity (LTPA)*			
Walking (min/week)	<150	675	86.6
	≥150	104	13.4
Total LTPA (min/week)	<150	586	75.2
	≥150	193	24.8
PA counseling received by a health professional	No	444	57.0
	Yes	335	43.0

PA: physical activity.

**Table 3 ijerph-18-05079-t003:** Associated factors with physical activity counseling among adults in Primary Health Care units. São José dos Pinhais, Paraná, Southern Brazil, 2019 (*n* = 779).

Variables	Category	Prevalence of PA Counseling (%)	Bivariate Analysis		Multivariate Analysis	
			PR (95% CI)	*p*	PR (95% CI)	*p*
*Level 1—Sociodemographic characteristics*						
Sex	Female	43.6	1			
	Male	41.7	0.95 (0.80–1.14)	0.632	-	-
Age group (years)	18–39	33.2	1		1 ^a^	
	40–59	50.9	1.53 (1.26–1.84)	<0.001	1.44 (1.19–1.75)	<0.001
	≥60	53.3	1.60 (1.29–1.99)	<0.001	1.44 (1.14–1.83)	0.002
Marital status	Single	36.1	1		1 ^b^	
	Married	46.7	1.29 (1.08–1.55)	0.006	1.27 (1.07–1.59)	0.007
Skin color	White	41.9	1			
	Non-white	46.4	1.11 (0.93–1.32)	0.249	-	-
Education level	Elementary education or less	53.0	1		1 ^c^	
	Junior high school	38.8	0.73 (0.57–0.93)	0.011	0.82 (0.64–1.04)	0.109
	High school or more	38.2	0.72 (0.60–0.85)	<0.001	0.82 (0.68–0.99)	0.039
Economic level	Low	44.0	1		-	-
	High	40.6	0.92 (0.76–1.11)	0.400	-	-
*Level 2—Health conditions*						
Body Mass Index (kg/m^2^)	<24.9	32.7	1		1 ^d^	
	25–29.9	39.2	1.20 (0.95–1.50)	0.120	1.09 (0.87–1.37)	0.456
	≥30.0	59.1	1.81 (1.47–2.23)	<0.001	1.53 (1.23–1.90)	<0.001
Smoking	No	44.5	1		1 ^e^	
	Yes	34.5	0.78 (0.59–1.01)	0.059	0.85 (0.66–1.09)	0.205
Hypertension	No	36.7	1		1 ^f^	
	Yes	54.3	1.48 (1.26–1.73)	<0.001	0.94 (0.77–1.15)	0.549
Diabetes	No	40.3	1		1 ^g^	
	Yes	57.4	1.42 (1.18–1.70)	<0.001	0.91 (0.74–1.13)	0.384
Dyslipidemia	No	39.7	1		1 ^h^	
	Yes	60.5	1.52 (1.28–1.80)	<0.001	1.15 (0.94–1.40)	0.179
Coronary artery disease	No	42.0	1		1 ^i^	
	Yes	56.9	1.35 (1.05–1.74)	0.020	1.01 (0.78–1.30)	0.938
Number of chronic diseases	0	33.3	1		1 ^j^	
	1	51.5	1.55 (1.28–1.87)	<0.001	1.18 (0.94–1.47)	0.154
	≥2	59.5	1.79 (1.48–1.87)	<0.001	1.11 (0.83–1.47)	0.483
Number of medications	0	28.7	1		1 ^k^	
	1–3	57.4	2.00 (1.66–2.41)	<0.001	1.83 (1.47–2.27)	<0.001
	≥4	56.5	1.97 (1.57–2.46)	<0.001	1.66 (1.21–2.28)	0.002
*Level 3—Leisure-time physical activity (LTPA)*						
Walking (min/week)	<150	40.6	1		1 ^l^	
	≥150	58.7	1.45 (1.20–1.73)	<0.001	1.28 (1.06–1.54)	0.010
Total LTPA (min/week)	<150	40.8	1		1 ^l^	
	≥150	49.7	1.22 (1.02–1.44)	0.024	1.16 (0.98–1.37)	0.094

^a^ adjusted by marital status and education level. ^b^ adjusted by age group and education level. ^c^ adjusted by age group and marital status. ^d^ adjusted by age group, marital status, education level, smoking, hypertension, diabetes, dyslipidemia, coronary artery disease, and number of medications. ^e^ adjusted by age group, marital status, education level, BMI, hypertension, diabetes, dyslipidemia, coronary artery disease, and number of medications. ^f^ adjusted by age group, marital status, education level, BMI, smoking, diabetes, dyslipidemia, coronary artery disease, and number of medications. ^g^ adjusted by age group, marital status, education level, BMI, smoking, hypertension, dyslipidemia, coronary artery disease, and number of medications. ^h^ adjusted by age group, marital status, education level, BMI, smoking, hypertension, diabetes, coronary artery disease, and number of medications. ^I^ adjusted by age group, marital status, education level, BMI, smoking, hypertension, diabetes, dyslipidemia, and number of medications. ^j^ adjusted by age group, marital status, education level, BMI, smoking, and number of medications. ^k^ adjusted by age group, marital status, education level, BMI, smoking, hypertension, diabetes, dyslipidemia, coronary artery disease. ^l^ adjusted by age group, marital status, education level, BMI, smoking, hypertension, diabetes, dyslipidemia, coronary artery disease, and number of medications.

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
