# Peer review of "Physical Activity Counseling among Adults in Primary Health Care Centers in Brazil"

_ijerph, 2021, doi:10.3390/ijerph18105079_

Round 1
Reviewer 1 Report
This article presents a cross sectional study to assess the counseling for Physical activity that occurs in Brazilian primary care. The problem with this manuscript is that is is not novel and it does not tell readers why this should be interesting to them. THere are countless articles in the published literature on counseling in primary care. for PA. why would this article be different and why Brazil?
Methods are sound an analyses are sound.
Author Response
We appreciate the reviewer's considerations. Our answers are in the attached file.

Reviewer 2 Report
This interesting, well designed, implemented and reported study meets the basic ethic requirements to be published.
However, some complements are needed:
1° the power calculated at 92% with a beta risk of 20% has to be made more explicit.
2° the level of incomes considered have to be reported.
3° 67% of the population are overweighted or obese persons; it would be interesting to report the number of persons with a BMI > 25 and <27, as a lot of authors consider overweight from a 27 BMI.
4°the number of smokers is very low (14.9%); this has to be discussed as it could be a selection bias of very health proactive persons.
5° more practical conclusions could be drawn with quantitative relevant aims.
Author Response

(The authors gave the same response as above.)

Reviewer 3 Report
The study is well planned, seriously done, interesting, original and adds to the literature. However, there are some concerns that need to be addressed before the manuscript could be considered for publication in the Journal of Environmental Research and Public Health.
Minor concerns:
- I would like to suggest to the authors to correct the table 2 and 3, there is a misprint.
- In Discussion please provide accurate or additional information about strengths of the study. What are the "instruments measured variables"? What do the authors mean with "valid and standardized procedures and instruments measured variables"? This point should be clarified.
Author Response

(The authors gave the same response as above.)

Round 2
Reviewer 1 Report
This is improved with the responses to the early reviewers. I have no more concerns
Reviewer 2 Report
REVIEWER'S OBSERVATIONS HAVE BEEN TAKEN INTO ACCOUNT.